# Sleep decreases neuronal activity control of microglial dynamics in mice

I. Hristovska[1,2,6], M. Robert [1,2,3,6], K. Combet [1,2,6], J. Honnorat[1,2,3], J-C Comte[2,4,5] & O. Pascual [1,2] ✉

Microglia, the brain-resident immune cells, are highly ramified with dynamic processes transiently contacting synapses. These contacts have been reported to be activity-dependent, but this has not been thoroughly studied yet, especially in physiological conditions. Here we investigate neuron-microglia contacts and microglia morphodynamics in mice in an activity-dependent context such as the vigilance states. We report that microglial morphodynamics and microglia-spine contacts are regulated by spontaneous and evoked neuronal activity. We also found that sleep modulates microglial morphodynamics through Cx3cr1 signaling. At the synaptic level, microglial processes are attracted towards active spines during wake, and this relationship is hindered during sleep. Finally, microglial contact increases spine activity, mainly during NREM sleep. Altogether, these results indicate that microglial function at synapses is dependent on neuronal activity and the vigilance states, providing evidence that microglia could be important for synaptic homeostasis and plasticity.

Microglial cells are the resident immune cells of the central nervous system. In the healthy brain, they perform many physiological tasks essential for neuronal homeostasis and synaptic functions. During development, they participate in shaping neuronal connectivity by phagocytic and non-phagocytic refinement of neuronal circuits, selective synaptic strengthening, and filopodia formation[1–5]. In the adult brain, microglia play a central role in learning-induced synapse formation by the secretion of brain-derived neurotrophic factor[6]. Moreover, they may enhance synaptic activity, maintain brain network synchronization within a physiological range and prevent hyperexcitability[7–9]. These synaptic functions are critically dependent on microglial morphodynamics, which refers to the changes in microglial morphology and motility. The regulation and functional properties of microglia-synapse interactions are a subject of intense research. Several studies have reported that global and local increase in neuronal activity leads to enhanced process motility, with process extension towards active sites and increased contact duration with neuronal elements[7,8,10–13]. The best-described mechanism involves microglial P2Y12R-mediated process extension, driven by activity-dependent ATP/ADP release at sites of increased activity[10,14]. Other mechanisms modulating microglial dynamics may include changes in potassium conductance, Cx3cr1/fractalkine signaling, as well as neuromodulator release, in particular norepinephrine and serotonin[15–19]. However, decreases in neuronal activity by sensory deprivation, optogenetic and pharmacological inhibition of neuronal activity have yielded inconsistent results on microglial dynamics and microglia-neuron interactions potentially due to the model and methods used and the activation state of microglia[11,17,18,20–23]. Nevertheless, stimulating or inhibiting neurons results in distinct changes in microglial gene expression and Ca$^{2+}$ activity in microglial processes, both in favor of microglial sensing of neuronal activity[9,24].

Major changes in neuronal activity in physiological conditions occur during the sleep-wake cycles. In addition, vigilance states are associated with distinct neuromodulation and ion/purine

[1]INSERM U1314, CNRS UMR5284, MeLiS, Lyon, France. [2]Université Claude Bernard Lyon 1, Lyon, France. [3]French Reference Center on Paraneoplastic Neurological Syndromes and Autoimmune Encephalitis, Hospices Civils de Lyon, Hôpital Neurologique, 59 Boulevard Pinel, 69677 Bron, Cedex, France. [4]INSERM U1028, CNRS UMR5292, Lyon, France. [5]Centre de Recherche en Neuroscience de Lyon, Lyon, France. [6]These authors contributed equally: I. Hristovska, M. Robert, K. Combet. ✉e-mail: olivier.pascual@inserm.fr

concentrations that may impact microglial morphodynamics[25–27]. Despite evidence for microglial sensing of neuronal activity and synaptic pruning in physiological conditions, the impact of the sleep-wake cycle on microglial dynamics and functional properties has not been elucidated yet. In awake mice, noradrenergic tone was recently found to suppress microglial process area and surveillance territory when compared to anesthesia[17,18], but a clear description of the microglia process dynamics during sleep and the underlying regulation is lacking. Although the exact function of sleep is still elusive, there is a strong consensus as to its role in synaptic homeostasis and plasticity. Indeed, sleep and wake are important for learning and memory consolidation and are associated with molecular mechanisms of synaptic plasticity that may involve microglial cells[28,29].

To better understand microglial morphodynamics in physiological conditions at the network and synaptic level, we combined electrophysiological recordings in vigil head-restrained mice with two-photon in vivo imaging of microglial cells and neuronal activity. Our results show that global microglial motility and complexity are overall correlated with changes in neuronal activity and that change in activity at the spine level impacts the proximity and contacts of microglia processes with synapses. These relationships between neuronal activity and microglia morphodynamics were also found during the alternance of wake and sleep and are dependent on Cx3cr1 signaling. At a single-spine level, microglial processes are attracted towards active spines during wake, whereas microglial proximity and contact with spines are downregulated during NREM sleep, in a state-dependent manner. Finally, we found that microglial contact with spines resulted in an increase in spine activity which was mainly observed during NREM sleep.

## Results

### Microglial morphodynamics and microglia-spine contacts are modulated by spontaneous neuronal activity

Microglia are constantly surveying the brain parenchyma, and changes in neuronal activity may impact their morphodynamics. However, the influence of neuronal network activity is not completely understood. For this reason, we assessed microglial morphodynamics in the somatosensory cortex of *Cx3cr1*[+/–] mice using two-photon microscopy while simultaneously monitoring brain activity with electroencephalogram (EEG) recordings (Fig. 1a). To this aim, we concurrently performed thinned-skull cortical window preparation and electroencephalogram (EEG)/electromyogram (EMG) electrode implantation (Fig. 1b). Brain activity was recorded and monitored during the 30–35 min of the imaging sessions. The activity was characterized by periods of high- and low-amplitude activity (Supplementary Fig. 2a). To study the relationship between microglial morphodynamics (Fig. 1c) and neuronal activity, we first performed the cross-correlation between microglial complexity (ramification) and EEG power, as well as the cross-correlation between motility (dynamics) and EEG power. We found that both microglial complexity (Fig. 1d) and motility (Fig. 1e) were negatively cross-correlated with the EEG power, suggesting that high-amplitude activity is linked to lower microglial morphodynamics and vice versa. This result suggests that microglia morphodynamics could be driven by global cortical activity. To assess this hypothesis, we calculated the correlation between the motility and complexity of pairs of microglial cells as a function of their inter-distance (Supplementary Fig. 2b). Our rationale was as follows: when microglial morphodynamics is synchronized for several microglial cells in our field of view (200 × 200 μm), this suggests a global impact of neuronal activity. On the other hand, if individual microglia behave independently from each other, a local regulation of motility and complexity might be conceivable. We found that the complexity changes between microglial cells are correlated (Fig. 1f), even though the correlation slightly decreases with the distance, from 0.84 when microglia are at a distance of 42 μm from each other to 0.25 when

separated from 98 μm. This result indicates that complexity might reflect a dependency on global brain activity. On the other hand, motility changes were weakly correlated with the EEG power, and the correlation strongly decreases when the distance between microglia increases (Fig. 1g) from 0.72 when microglia are at a distance of 42 μm from each other to −0.31 when microglia are at a distance of 98 μm (Supplementary Fig. 2b). These results suggest that the regulation of microglial complexity and microglial motility originate from different levels of integration. While complexity seems to respond to global neuronal activity, motility rather depends on the local network, possibly at the synaptic level.

Previous studies have found that microglial processes are in close proximity with spines, and that this relationship may be activity-dependent[8,11,18]. To study microglial dynamics at the synaptic level, we simultaneously recorded microglial cells in *Cx3cr1CreERT2*[+/–]*ROSA26-STOP-tdTomato*[–/–] mice, expressing the reporter gene tdTomato in microglia, and fluctuations of intracellular $Ca^{2+}$ concentration as a proxy of neuronal activity at the dendritic spine level after GCaMP6 injection (Fig. 1b). We found that microglial processes make frequent contacts with dendritic spines (Fig. 2a) lasting on average ~4.38 min ± 0.42 (Supplementary Fig. 2c). Active spines were contacted on average ~35% of the time by microglial processes, even though contact duration was highly variable (Supplementary Fig. 2d).

To assess whether local spine activity affects microglia-spine interaction, we studied the correlation between the activity of the spine and the duration of microglia-spine contact. We found that this correlation was positive and that microglial processes stayed longer in contact with highly active spines (Fig. 2b, Pearson's correlation test, *p* < 0.05). To test whether this profile may be observed by a random distribution of values for contact duration and spine activity, we performed the same analysis with scrambled original data. We did not find any correlation for this random data set (Fig. 2c, Pearson's correlation test, *p* > 0.05), indicating that microglial contact with spines is activity-dependent.

### Modulation of neuronal activity by whisker stimulations changes microglial process proximity and contacts with spines

To strengthen our observations and provide a causal evidence of the relationship between microglial dynamics and neuronal activity, we decided to modulate neuronal activity by stimulating the C2 whisker (Fig. 3a) while monitoring spine activity (Fig. 3b) and microglial processes in the corresponding contralateral barrel. We first ensured that the repetition of stimulations (1-s stimulation at 90 Hz every 30 s during 25 min) did not induce synaptic plasticity of the network (Supplementary Fig. 3a, one-way ANOVA, *p* > 0.05). Because of the polysynaptic nature of the network, we found that stimulations of the whisker induced three different types of response (Supplementary Fig. 3b). 39.3% of the spines studied responded by an increase of activity (Fig. 3c–e, unpaired t-test, ****p* < 0.0001; e, paired t-test, two-tailed, *****p* < 0.0001), 24.9% of spines by a decrease of activity (Fig. 3c, d, f, unpaired t-test, *****p* < 0.0001; f, paired t-test, two-tailed, *****p* < 0.0001) while 35.8% of spines did not respond significantly to whisker stimulations (Supplementary Fig. 3c, d, one-way ANOVA, *****p* < 0.0001; d, paired t-test, two-tailed, *p* > 0.05). We found that the contact duration between microglial processes and spines was longer for the responding spines compared to decreased activity spines during stimulations (Fig. 3g, unpaired t-test, *p* < 0.05) while not being different during baseline (Supplementary Fig. 3e, unpaired t-test, *p* > 0.05). For the 147 spines that responded by an increased activity we found that the proportion of spines contacted by a microglial process at a given time increased by 7.9 ± 3.2 % after 10-15 min stimulation when compared to baseline (Fig. 3h, one-way ANOVA, *p* < 0.05) and the spines spent on average 12.7 ± 4.4% more time after 10–15 min of whisker stimulations (Fig. 3i, one-way ANOVA, *p* < 0.05 **p* < 0.001). On the contrary, for the 93 spines that exhibited a decreased activity

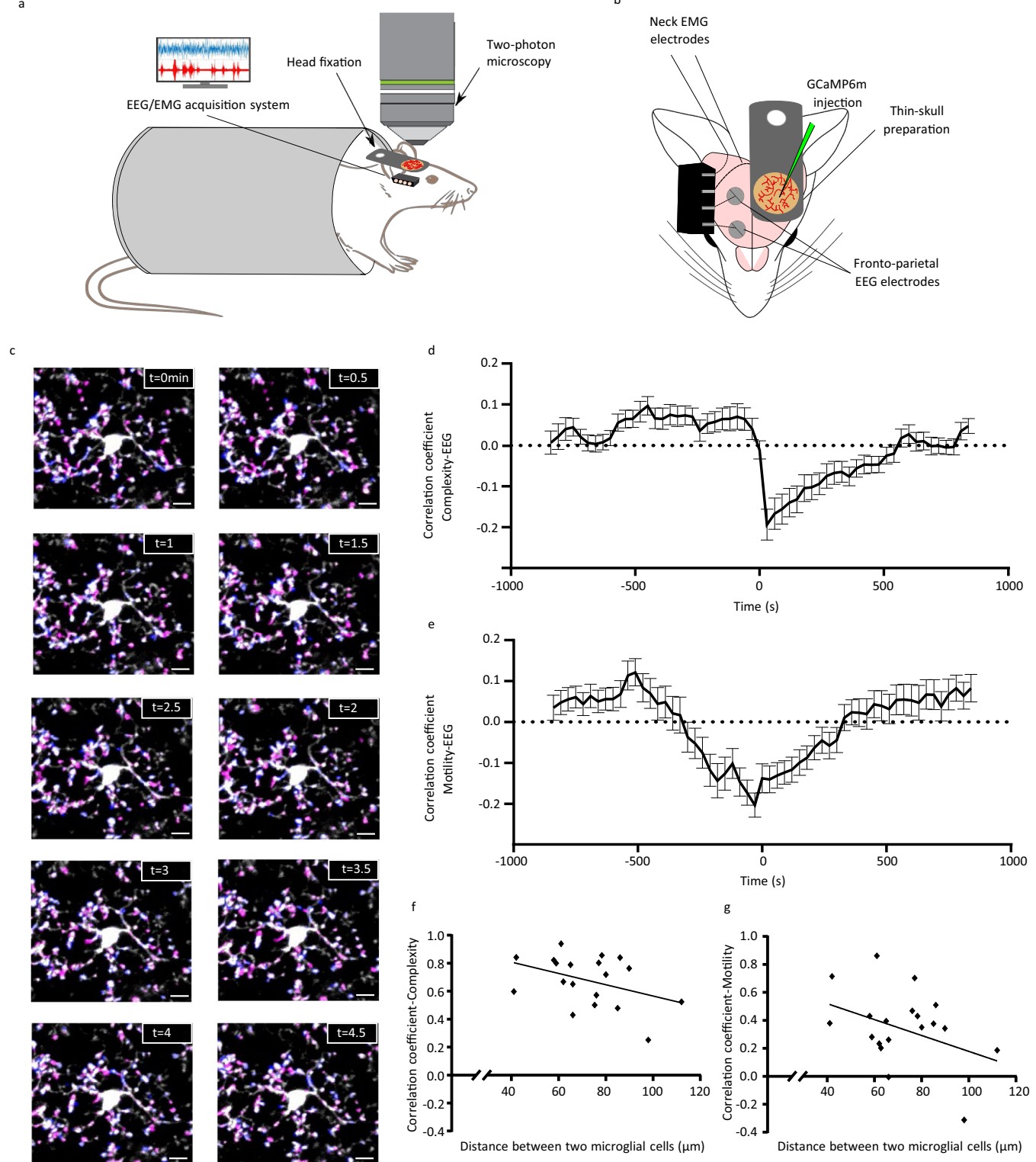

**Fig. 1 | Neuronal activity correlates with microglial complexity and motility.**
**a, b** Schematic representation of our experimental setup. **a** Head-restrained mice were trained for simultaneous two-photon imaging with electrophysiology (side view). **b** Fronto-parietal EEG and neck EMG electrodes were implanted, combined with a thinned-skull cortical window preparation and calcium indicator injection (top view). **c** Example of microglial motility over a 5-min period, showing microglial process retractions in magenta, and process extensions in cyan. Scale bar = 10 µm. Cross-correlation analysis between **d** microglial complexity ($n = 6$ mice, 2–7 microglial cells/mouse) and **e** motility ($n = 6$ mice, 2–7 microglial cells/mouse) with EEG power. This experiment was repeated independently with similar results, as shown in Fig. 5h-i. Distribution of the correlation coefficient of the **f** complexity ($n = 3$–4 pairs of microglial cells per mouse) and **g** motility ($n = 3$–4 pairs of microglial cells per mouse) for pairs of microglial cells as a function of their distance. Source data are provided as a Source Data file for **d**–**g**.

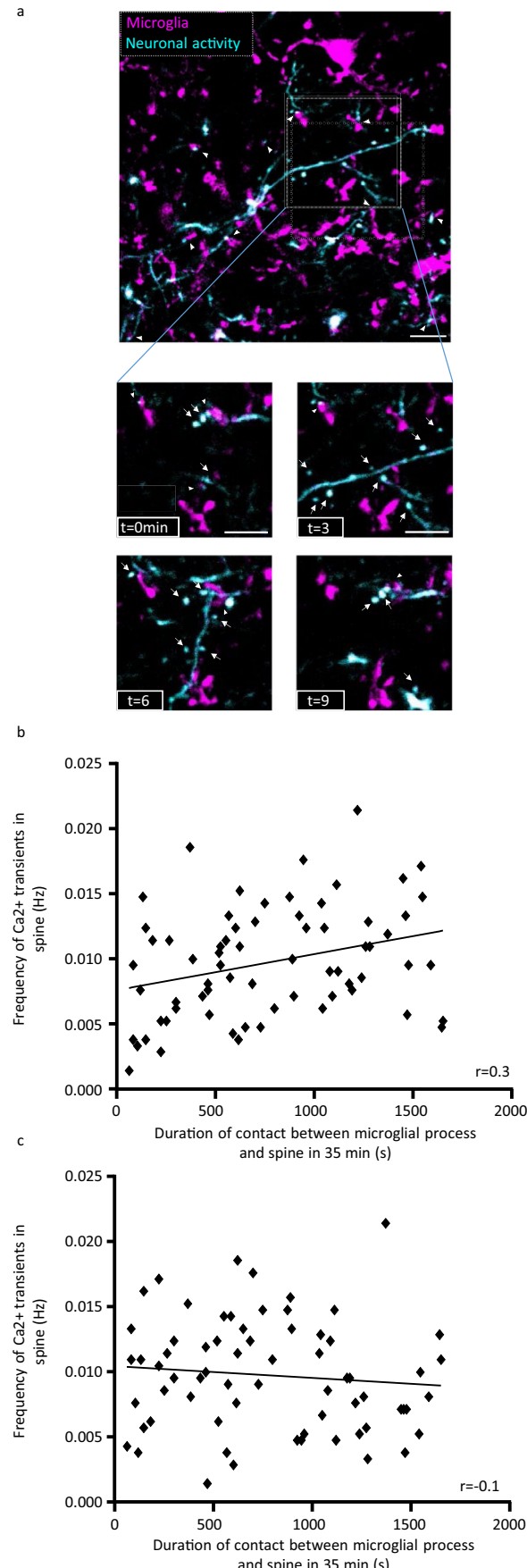

**Fig. 2 | Microglia contact spines in an activity-dependent manner.**
**a** Simultaneous imaging of microglial processes (magenta, arrow) and neuronal GCaMP6 expression (cyan) in the L1 of the primary somatosensory cortex. The arrowheads indicate contacts between microglial processes and spines. The dynamic interactions between microglial processes and spines are shown in a smaller FOV (delimited by the white square) over a 10-min period. Examples of microglial processes interacting with spines are indicated with the white arrowheads, whereas spines not in contact with microglia are pointed with white arrows. The brightness and contrast were adjusted for visualization purposes. One representative image with ten replicates. **b** Correlation between Ca$^{2+}$ spine frequency and the duration of microglia-spine contact ($n = 68$ spines from five mice, Pearson's correlation test, two-tailed, $p = 0.012$). **c** Correlation between spine activity and duration of microglia-spine contact for a random data set ($n = 68$ spines from five mice, Pearson's correlation test, two-tailed, $p = 0.42$). All data are represented as mean ± SEM. Source data are provided as a Source Data file for **b**, **c**.

during simulations, we found that the proportion of spines in contact during stimulation (Fig. 3j, one-way ANOVA, *$p < 0.05$) as well as the percentage of time that spines spent in contact with microglial processes decreased by 11.2 ± 4.8% and 13.3 ± 5.8%, respectively 10–15 min following the onset of the stimulation (Fig. 3k, one-way ANOVA, *$p < 0.05$). Finally, when measuring the relative distance of the closest microglial process with respect to spines, we found that microglial processes were closer during stimulation for spines that responded by an increase of activity and further away during stimulation for spines that responded by a decrease of activity (Fig. 3l, Kolmogorov-Smirnov test, **$p < 0.01$), while no difference was observed between the groups during the baseline period (Supplementary Fig. 3f, Kolmogorov-Smirnov test, $p > 0.05$). It is interesting to note that for spines which did not exhibit a response to the stimulations, no change was observed in the proportion of spines being contacted, the percentage of time spent in contact, and the distance between the closest microglial process and the spine (Supplementary Fig. 3g–i, one-way ANOVA, $p > 0.05$; h, one-way ANOVA, $p > 0.05$; i, Kolmogorov-Smirnov test, $p > 0.05$). Altogether these results indicate that modulation of neuronal activity induces changes in the proximity of microglial processes with respect to spines, the proportion of spines being contacted by a microglial process, and the contact duration.

## Vigilance states influence microglial morphodynamics and microglia-spine interaction

The vigilance states are physiological conditions during which activity is drastically changing. To examine whether microglial complexity and motility are influenced by the change of neuronal activity associated with the vigilance states, we performed two-photon imaging of microglial cells while simultaneously scoring the vigilance states through EEG and EMG recordings. Each session included several episodes of wake and NREM sleep (Fig. 4a, b) during which microglial morphodynamics varied (Supplementary Fig. 4a, b). The same microglial cell imaged during wake and NREM sleep displayed higher motility and complexity (wake in magenta and sleep in cyan) during wake compared to NREM sleep (Fig. 4c and Supplementary Fig. 4c). Quantifications of microglial morphology and motility in vivo revealed that microglial complexity (Fig. 4d, Wilcoxon test, *$p < 0.05$) and motility (Fig. 4e, Wilcoxon test, *$p < 0.05$) were significantly reduced during NREM sleep compared to wake. Of note, the average proportions of NREM sleep and wake were comparable for the recordings chosen for analysis (Supplementary Fig. 4d, Wilcoxon test, $p > 0.05$), and one episode of a given vigilance state lasted on average 5.86 min ± 0.46. Further details about the characteristics of the sleep-wake episodes in our recordings can be found in Supplementary Figs. 4e–l.

These results indicate that microglia are sensing the vigilance states, possibly through the change of neuronal activity associated with a given vigilance state, and adapt their morphodynamics accordingly.

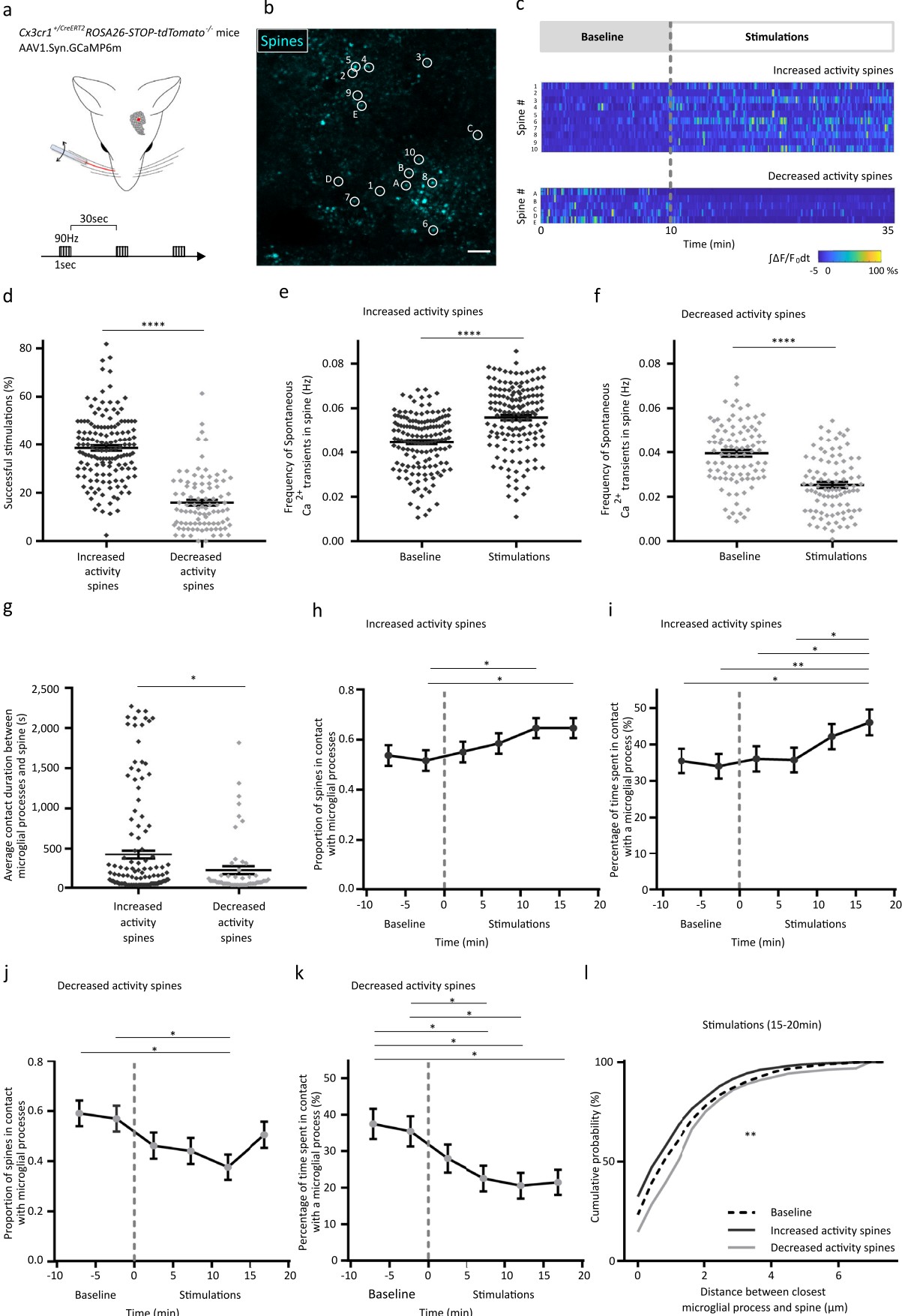

**Fig. 3 | Modulation of neuronal activity by whisker stimulation controls microglial dynamics towards dendritic spines. a** Experimental set-up: Contralateral single whisker stimulation (90 Hz, 1 s) every 30 s, relative to the imaging location in the corresponding somatosensory barrel cortex in *Cx3cr1CreERT2*[+/-] *ROSA26-STOP-tdTomato*[-/-] mice transduced with AAV1.Syn.GCaMP6m. **b** Example of one image from 9 replicates, showing maximum intensity time and Z-projections of the Ca²⁺ signal (35 minutes, 14 planes, 1 μm/plane) used for spine selection. Scale bar=10 μm. Brightness and contrast were adjusted for visualization purposes. **c** Normalized color-coded calcium activity of a set of 15 individual spines shown in **b**, categorized as Increased and Decreased activity spines. **d** Average success rate of individual spines in response to the stimulations ($n_{IA}$ = 147 spines and $n_{DA}$ = 93 spines from three mice, unpaired t-test, two-tailed, ****$p < 0.0001$). Average spontaneous frequency of Ca²⁺ events for individual spines plotted before and during the stimulation period for **e** Increased activity spines ($n_{IA}$ = 147 spines from three mice, paired t-test, two-tailed, ****$p < 0.0001$) and **f** Decreased activity spines ($n_{DA}$ = 93 spines from three mice, paired t-test, two-tailed, ****$p < 0.0001$). **g** Average duration of microglia-spine contact during the stimulations ($n_{IA}$ = 141

contacts and $n_{DA}$ = 57 contacts from three mice, unpaired t-test, two-tailed, *$p = 0.028$). **h** Kinetics of the proportion of spines contacted by microglial processes for Increased activity spines ($n_{IA}$ = 147 spines from three mice, one-way ANOVA, *$p = 0.048$) and **i** Average percentage of time individual spines spent in contact with microglial processes for Increased activity spines ($n_{IA}$ = 147 spines from 3 mice, one-way ANOVA, *$p = 0.0396$, **$p = 0.0063$, *$p = 0.0394$ and *$p = 0.0133$). **j** Kinetics of the proportion of spines contacted by microglial processes for Decreased activity spines ($n_{DA}$ = 93 spines from three mice, one-way ANOVA, *$p = 0.0255$ and *$p = 0.02$) and **k** Average percentage of time individual spines spent in contact with microglial processes for Decreased activity spines ($n_{DA}$ = 93 spines from three mice, one-way ANOVA, *$p = 0.027$, *$p = 0.0118$, *$p = 0.0206$, *$p = 0.0479$ and *$p = 0.0451$). **l** Cumulative probability of the distance between the closest microglial process during a 5-min episode during the stimulation period ($n_{IA}$ = 147 spines and $n_{DA}$ = 93 spines from three mice, Kolmogorov-Smirnov test, **$p < 0.002$). All data are represented as mean ± SEM. Source data are provided as a Source Data file for **d**–**l**.

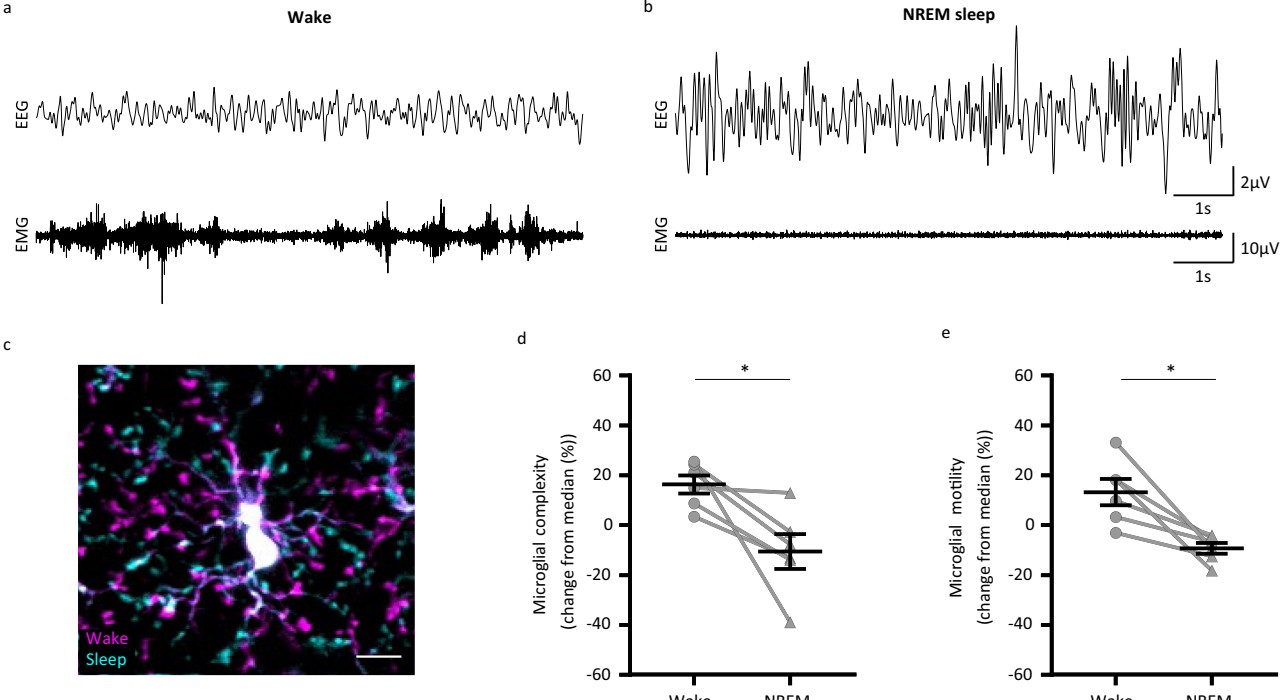

**Fig. 4 | Microglial morphodynamics is modulated by the vigilance states.** Representative EEG and EMG traces during **a** wake and **b** sleep. **c** A color-coded image showing microglial complexity during wake (in magenta) and during sleep (in cyan). Scale bar = 10 μm. Quantification of **d** microglial complexity ($n$ = 6 mice, 3–7 microglial cells/mouse, Wilcoxon test, two-tailed, *$p = 0.0313$) and **e** motility

($n$ = 6 mice, 3–7 microglial cells/mouse, Wilcoxon test, two-tailed, *$p = 0.031$) during wake and sleep. This experiment was repeated independently with similar results, as shown in Fig. 5b, c. All data are represented as mean ± SEM. Source data are provided as a Source Data file for **d**, **e**.

## The Cx3cr1 signaling is involved in the change of microglial morphodynamics associated with the vigilance states

Since Cx3cr1 has been shown to be involved in the motility of microglial cells[16,30], we next investigated whether the invalidation of the *Cx3cr1* gene could affect the morphodynamic changes associated with the vigilance states. To do this, we performed *in vivo* two-photon imaging experiments using *Cx3cr1*[eGFP/eGFP] mice in which *Cx3cr1* was knock out on both alleles (*Cx3cr1*[-/-]) mice that we compared to *Cx3cr1*[+/eGFP] that we will refer to later on as *Cx3cr1*[+/-] in the text. When monitoring microglial motility and complexity during the alternation of the vigilance states, we found that microglia from *Cx3cr1*[-/-] mice did not exhibit changes of complexity nor motility associated with the shift of vigilance states as it was the case for *Cx3cr1*[+/-] mice (Fig. 5a–c, Wilcoxon test, *$p < 0.05$). This result suggests that Cx3cr1 is involved in the regulation of microglial

morphodynamics associated with the vigilance states. Because the deletion of the *Cx3cr1* gene in microglia could alter the quality of sleep, we analyzed and compared sleep episodes between *Cx3cr1*[-/-] and *Cx3cr1*[+/-] mice. Overall, the percentage of time spent in NREM sleep, the duration, and the number of NREM sleep episodes were not significantly different between groups (Fig. 5d–f, Mann-Whitney test, *$p < 0.05$). Another possibility explaining the absence of motility change in homozygous mice could be that the neuronal activity differs from the one monitored in heterozygous mice. To test that hypothesis, we compared the EEG power spectrum of *Cx3cr1*[-/-] and *Cx3cr1*[+/-] mice and found no differences in the amplitude-frequency plots (Fig. 5g, unpaired t-test, two-tailed, $p > 0.05$), indicating that global cortical activity was similar between *Cx3cr1*[+/-] and *Cx3cr1*[-/-] mice. Finally, we looked at the cross-correlation between EEG power and microglial complexity or

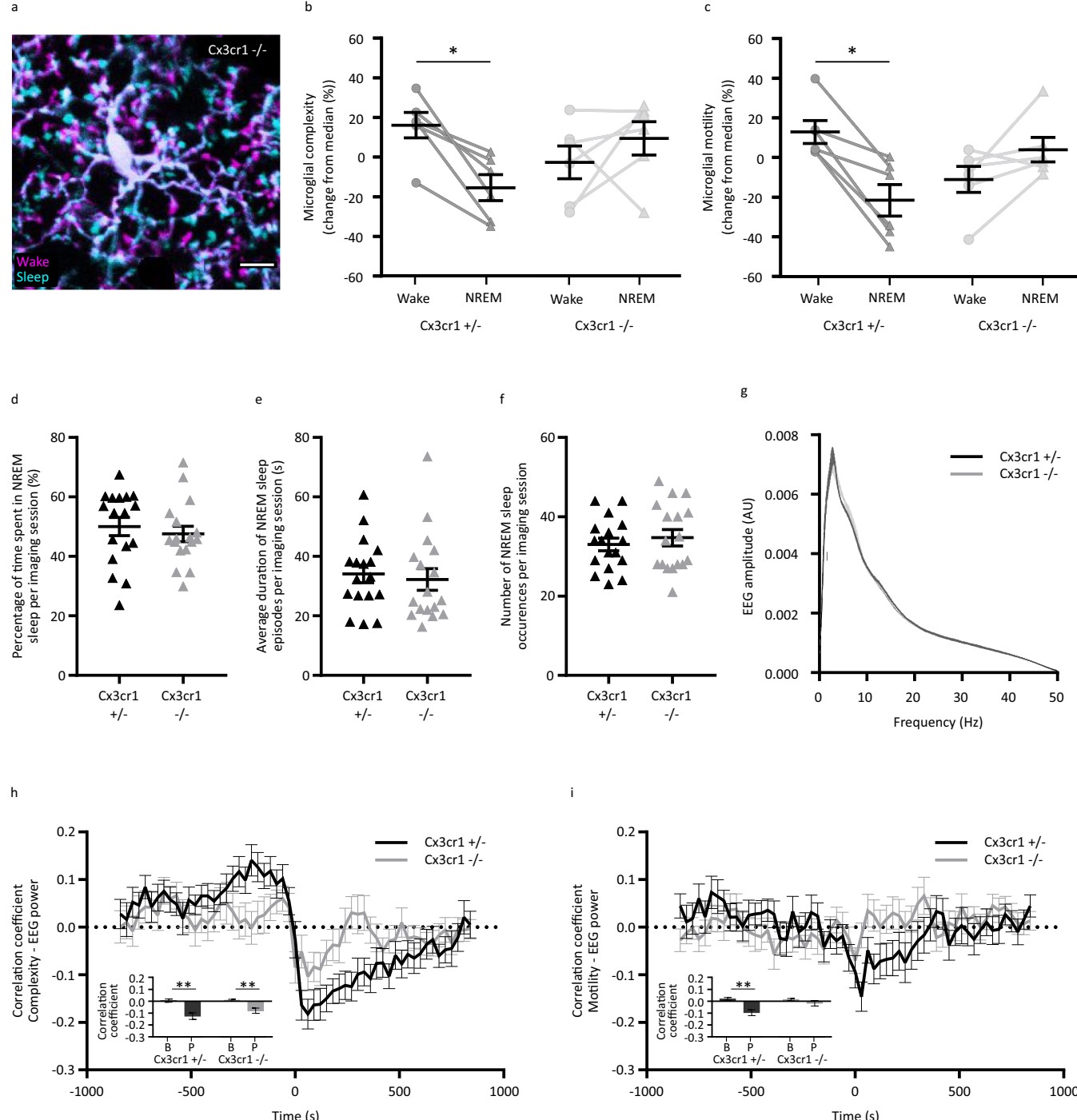

**Fig. 5 | Cx3cr1 is involved in the microglial morphodynamics changes during sleep/wake cycles. a** Selected frames showing one microglial cell from a *Cx3cr1$^{-/-}$* mouse during wake (magenta) and sleep (cyan). Scale bar = 10 μm. Quantification of **b** microglial complexity in *Cx3cr1$^{+/-}$* and *Cx3cr1$^{-/-}$* mice ($n$ = 6 mice, Wilcoxon test, two-tailed, *$p$ = 0.0313 and $p$ = 0.4375), and **c** microglial motility in *Cx3cr1$^{+/-}$* and *Cx3cr1$^{-/-}$* mice ($n$ = 6 mice, Wilcoxon test, two-tailed, *$p$ = 0.0313 and $p$ = 0.1563) during wake and sleep episodes. Quantification of NREM sleep during a 35-min imaging session between *Cx3cr1$^{+/-}$* and *Cx3cr1$^{-/-}$* mice through evaluation of **d** percentage of time, **e** average duration of individual episodes and **f** total number of episodes

($n$ = 17 sessions from six mice for both *Cx3cr1$^{+/-}$* and *Cx3cr1$^{-/-}$* mice, Mann-Whitney test, two-tailed, $p$ = 0.3891, p = 0.4482 and $p$ = 0.6762 respectively). **g** Comparison of EEG amplitude-frequency plot of *Cx3cr1$^{+/-}$* (black, $n$ = 6 mice) and *Cx3cr1$^{-/-}$* (gray, $n$ = 4 mice) mice. Cross-correlation analysis between **h** average microglial complexity and **i** average microglial motility with regards to the average EEG power ($n$ = 20 cells for *Cx3cr1$^{+/-}$* and $n$ = 22 cells for *Cx3cr1$^{-/-}$* respectively, Wilcoxon test, two-tailed, **$p$ = 0.0023 and **$p$ = 0.0042 for complexity and **$p$ = 0.0032 and $p$ = 0.4060 for motility, for *Cx3cr1$^{+/-}$* and *Cx3cr1$^{-/-}$* respectively). All data are represented as mean ± SEM. Source data are provided as a Source Data file for **b**–**i**.

motility and we found that the correlation was weaker in *Cx3cr1$^{-/-}$* when compared to *Cx3cr1$^{+/-}$* mice (Fig. 5h, i, Wilcoxon test, *$p$ < 0.05). This indicates that the relationship between neuronal activity and microglial morphodynamics is lost when the Cx3cr1

receptor is absent. Altogether these results indicate that the Cx3cr1 signaling is important in the regulation of microglial cells' morphodynamics with regards to neuronal activity.

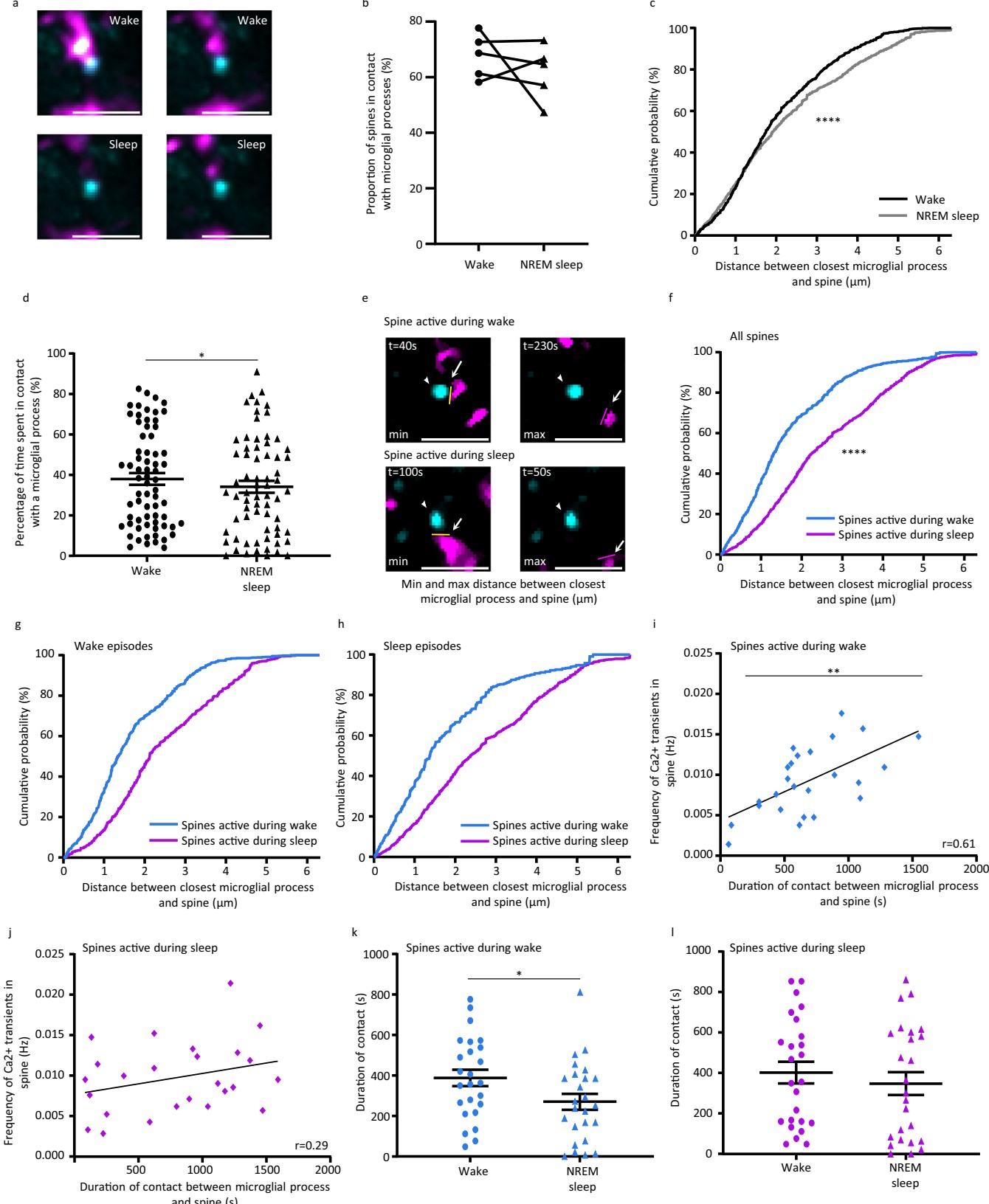

## NREM sleep limits the attraction of microglial processes toward active spines

Since microglial processes appear to sense neuronal activity and are attracted toward active spines, we wanted to assess whether this attraction may be influenced by the vigilance states. We first found that

the frequency of spine activity is not significantly different during wake and NREM sleep (Supplementary Fig. 5a, paired t-test, two-tailed, $p > 0.05$, Supplementary Fig. 5b for results per mice). Microglial processes appeared closer to spines and their contact duration with spines was longer during episodes of wake when compared to sleep (Fig. 6a).

**Fig. 6 | Activity during wake exerts positive attraction towards microglial processes. a** Color-coded cumulative projection of microglial processes around one spine over 4.2-min episodes of wake and sleep showing microglial distribution with regards to spines during the vigilance stages. The brightness and contrast were adjusted for visualization purposes. Scale bar = 10 μm. Representative images from the analysis present in Fig. 6c, replicated for >15 episodes of wake or sleep. **b** Proportion of contacted spines by microglial processes during individual 4.2-min wake and sleep episodes (*n* = 68 spines from five mice, Wilcoxon test, two-tailed, p = 0.625). **c** Cumulative distribution of the distance between the closest microglial process with spines during wake and sleep episodes (Kolmogorov-Smirnov test, two-tailed, ****p < 0.0001). **d** Percentage of time spent by the spine in contact with a microglial process during wake and sleep (*n* = 68 spines from five mice, paired t-test, two-tailed, *p = 0.047). **e** Selected frames showing the minimal distances (yellow line) and maximal distances (purple lines) between the closest microglial processes and a spine active during wake (upper panel) and a spine active during sleep (lower panel). The brightness and contrast were adjusted for visualization purposes. Scale bar = 10 μm. Representative images from the analysis present in

Fig. 6c, replicated 25 times for spines active during wake and spines active during sleep. **f** The cumulative distribution of microglia-spine distance for spines active during wake (blue line) and spines active during sleep (magenta line) (Kolmogorov-Smirnov test, two-tailed, ****p < 0.0001). The cumulative distribution of microglia-spine distance for **g** spines active during wake and **h** spines active during sleep during wake and sleep episodes (Kolmogorov-Smirnov test, two-tailed, ****p < 0.0001). All distances are indicated in μm. Correlation between microglia-spine contact duration and spine activity during **i** wake and **j** sleep (*n* = 25 for spines active during wake and *n* = 25 for spines active during sleep, from five mice, Pearson's correlation test, two-tailed, **p = 0.0013 for spines active during wake, and p = 0.16 for spines active during sleep). For episodes of wake and sleep, duration of contact between microglial processes and **k** spines active during wake (*n* = 25 spines from five mice, paired t-test, two-tailed, *p = 0.0125) and **l** spines active during sleep (*n* = 25 spines from five mice, paired t-test, two-tailed, p = 0.232). All data are represented as mean ± SEM. Source data are provided as a Source Data file for **b**–**l**.

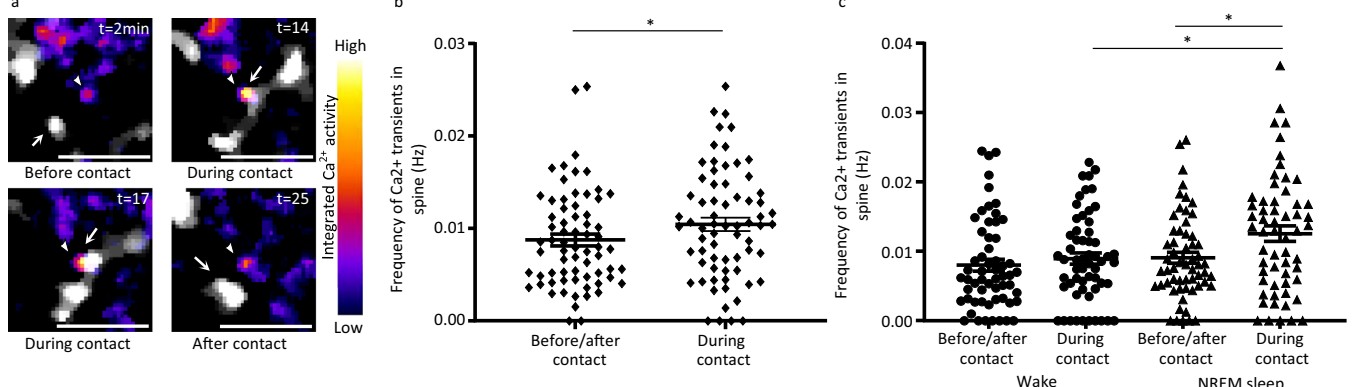

**Fig. 7 | Microglial contact with spines increases spine activity during sleep. a** Selected frames showing spine activity before/after and during contact with microglial processes. **b** Frequency of spine Ca²⁺ transients during or before/after contact with microglial processes (*n* = 68 spines from five mice, paired t-test, two-tailed, *p = 0.0326). **c** Frequency of spine Ca²⁺ before/after and during microglial contact during episodes of wake and sleep (*n* = 58 spines from five mice, one-way ANOVA, Tukey's multiple comparisons test; p > 0.05, except for comparison

between spine Ca²⁺ frequency during contact during episodes of wake vs episodes of sleep, *p = 0.044, comparison between spine Ca²⁺ frequency before/after contact during episodes of sleep vs spine Ca²⁺ frequency during contact during episodes of sleep, *p = 0.0218, and comparison between spine Ca²⁺ frequency before/after contact during episodes of wake vs spine Ca²⁺ frequency during contact during episodes of sleep, *p = 0.01). All data are represented as mean ± SEM. Source data are provided as a Source Data file for **b**, **c**.

We then quantified the proportion of spines in contact with microglial processes during wake and NREM sleep, and found no significant difference (Fig. 6b, Wilcoxon test, p > 0.05). However, by investigating the distribution of microglial processes around spines during the different vigilance states, we found that microglial processes were closer to spines during wake (Fig. 6c, Kolmogorov–Smirnov test, ****p < 0.0001). Furthermore, microglial processes spent slightly more time in contact with spines during wake when compared to NREM sleep (Fig. 6d, paired t-test, two-tailed, *p < 0.05, Supplementary Fig. 5c for results per mice), indicating an important influence of NREM sleep on microglia–spine interactions.

Having observed the impact of NREM sleep on the global distribution of processes and duration of microglia-spine contacts, we next wanted to evaluate more precisely the effect of NREM sleep on the positive attraction of microglial processes towards active spines. For this reason, we studied microglia-spine interactions in various regimens of synaptic activity during the vigilance states. We classified spines into three categories: spines mostly active during wake, spines mostly active during NREM sleep, and spines with balanced activity between NREM sleep and wake (Supplementary Fig. 1b), and we were interested in spines that exhibited most activity during one of the vigilance states. We studied the proximity of microglial processes with regards to spines mostly active during wake or during NREM sleep, and the duration of microglia-spine contact. We found that microglial

processes were closer to spines active during wake (Fig. 6e, f, Kolmogorov-Smirnov test, ****p < 0.0001). This observation holds true whatever the vigilance state during which the contact was monitored (Fig. 6g, h, Kolmogorov-Smirnov test, ****p < 0.0001). Furthermore, microglial processes were closer during wake both for spines mostly active during wake and spines mostly active during NREM sleep, suggesting that this relationship is impeded during NREM sleep (Supplementary Fig. 5b, c, Kolmogorov-Smirnov test, ****p < 0.0001). Spine activity was well correlated with microglia-spine contact duration for spines active during wake, but not as well for spines active during NREM sleep (Fig. 6i, j, Pearson's correlation test, **p < 0.01). In addition, the contact duration was reduced during NREM sleep for spines active during wake (Fig. 6k, paired t-test, two-tailed, *p < 0.05, Supplementary Fig. 5f for results per mice). This was not detected for spines active during NREM sleep (Fig. 6l, paired t-test, two-tailed, p > 0.05, Supplementary Fig. 5g for results per mice).

### Microglial contact induces spine activity increase during NREM sleep

Since microglial contact was recently found to increase spine activity[8], we wanted to assess whether this increase in activity may occur in both states or is vigilance state-dependent. By quantifying spine Ca²⁺ transients without and during contact with microglial processes, we confirmed that overall spine activity was significantly

increased during contact with microglial processes without considering the vigilance state (Fig. 7a, b, b, paired t-test, *$p < 0.05$, Supplementary Fig. 5h for results per mice). Intriguingly, during contact, the increase in spine activity was significantly higher during NREM sleep compared to wake (Fig. 7c, one-way ANOVA, *$p < 0.05$, Supplementary Fig. 5i for results per mice), indicating that the contact-induced increase in activity may be vigilance state-dependent, mainly occurring during NREM sleep.

## Discussion

In this work, we investigated the impact of neuronal activity and vigilance states on microglial dynamics at the network and the synaptic level. We report that neuronal activity largely contributes to changes in microglial morphodynamics. This was observed when monitoring spontaneous neuronal activity, as well as when neuronal activity was modulated by physiological stimulations. We also found that microglial morphodynamics was modulated by the vigilance states, likely mediated by the changes in neuronal activity, through a signaling pathway involving Cx3cr1. Microglial morphology was globally following the changes of neuronal activity associated with the vigilance states, whereas motility was likely regulated at the synaptic level. Finally, microglial contact increased spine activity, mostly during NREM sleep.

In physiological conditions, microglial processes are highly ramified and dynamic, continuously surveying the surrounding parenchyma, but their regulation remains poorly understood[22,23,31]. By concurrently monitoring microglial morphodynamics and the electrical activity of the brain in vigilant mice, we found that EEG power was negatively cross-correlated with microglial morphodynamics, linking neuronal activity patterns to microglial morphodynamics (Fig. 1d,e). High EEG power, reminiscent of synchronized activity and slow-wave dominated state, may be observed after administration of certain anesthetics, such as ketamine/xylazine or fentanyl cocktail. Several studies have assessed their impact on microglial morphodynamics, but results have been inconclusive, possibly due to differences in methodology, brain region studied, and quantification of morphodynamics parameters[17,18,20,21]. In a previous study, we found that ketamine/xylazine, dominated by high-amplitude, low-frequency oscillations, led to a reduction in microglial morphodynamics, suggesting important regulation of microglial morphology by the patterns of neuronal activity. An interesting question arising from this finding is whether the temporal evolution of microglial morphodynamics is comparable between microglial cells or is intrinsic to each cell. Changes in morphology were highly correlated for microglial cells (<120 μm apart), suggesting a global impact of neuronal activity on microglial morphology and its degree of ramification. On the other hand, motility changes were not as well correlated between microglial cells, indicating that motility could be regulated at the local level. Indeed, spontaneously active neurons, HFS, or trains of AP in a single neuron may recruit microglial processes potentially at the synaptic level[8,12,32]. In our study, we found that microglial processes spent more time in contact with spines displaying high $Ca^{2+}$ transient frequencies (Figs. 2, 3). This observation was true for spontaneous activity or when the network was recruited by physiological stimuli like whisker stimulations, demonstrating a causal link between neuronal activity and microglial contact with active spines. These results are in accordance with recent studies showing that microglia are preferentially in contact with active spines[8], and that processes extend their contacts with spines after LTP induction in hippocampal slices[12]. We decided not to use optogenetic or chemogenetic approaches because these are difficult to implement without being very invasive and thus without activating or priming microglial cells. Our study, along with previous findings, corroborate the notion that microglial processes may sense activity and may be attracted towards spines depending on their activity.

Taking into consideration the growing body of evidence of microglial guidance by neuronal activity and its potential involvement in synaptic plasticity, we proceeded in assessing the regulation of microglial morphodynamics by the vigilance states. Our findings indicate that NREM sleep is associated with a significant reduction of microglial morphological parameters and global motility (Fig. 4). Microglial complexity was reduced, pointing to a global decrease in microglial ramification during sleep. Even though sleep episodes were, as anticipated, shorter than those observed in freely moving animals[33], changes in neuronal activity characteristic of NREM sleep were monitored by EEG recordings, and changes in morphodynamics were observed during single episodes. Several episodes of REM sleep were detected, but they were too short and rare to be analyzed reliably.

Our study is the first to examine microglial morphodynamics during physiological sleep. Two recent studies have suggested that microglial cells exhibit a relatively low process surveillance in awake conditions, mediated by a high noradrenergic tone[17,18]. In more general terms, Liu et al. suggested that microglial dynamics follows a "U-shape" model, where significantly increasing or decreasing neuronal activity compared to wake state leads to increased microglial process dynamics. However, it is important to note that some of these results were obtained from a craniotomy which was found to profoundly affect spine turnover and microglial activation[34–36], whereas the results described in the current study are obtained using a thinned-skull preparation. It has also been shown recently that noradrenergic tone is high during NREM sleep especially during spindle episodes[37]. These latest results indicate that it is difficult to speculate on the role of noradrenaline during vigilance state in absence of proper measurements of the noradrenaline tone during the alternation of wake and sleep in the cortex. The mechanisms driving microglia morphodynamic changes are still elusive and are likely diverse. Excitatory and inhibitory neurotransmitters (NTs) fluctuate at the cortical level during the vigilance states[38]. Increase in ionotropic GABA transmission, found during sleep, resulted in significant reduction of microglial motility and morphology in retinal explants, which is in line with our results[39]. During wake, among other NTs, cortical norepinephrine and serotonin are increased[40,41], but these have divergent effects on microglial cells. Norepinephrine caused microglial process retraction[17,18,42], whereas serotonin has chemotactic effect on microglial processes[43,44]. Thus, the effect of the complex neuromodulatory environment of the sleep-wake cycles on microglial morphodynamics needs to be further assessed. Besides NTs, several ions exhibit state-dependent changes of concentration[26]; for instance arousal is associated with a rapid rise in $[K^+]_e$, along with a slower decrease in $[Ca^{2+}]_e$ and $[Mg^{2+}]_e$[26]. Baseline microglial morphodynamics is regulated by changes in $[K^+]_e$[15], however, the $[K^+]_e$ change required to induce microglial dynamics[15] is far beyond the slight $[K^+]_e$ shift[26] observed during the vigilance states. Therefore, changes occurring in the extracellular space during the vigilance states are multiple and the exact mechanisms influencing microglial morphodynamics remain to be determined.

Several studies have investigated the role of Cx3cr1 and various results were found depending on the area investigated. In the visual cortex, Cx3cr1 did not seem to be required for activity-dependent plasticity[45], while in the hippocampus Cx3cr1 was shown to be involved in synaptic functions[30,46]. In our study, we explored the involvement of Cx3cr1 signaling in microglia-neuron communication (Fig. 5). Our results indicate that when the membrane Cx3cr1 receptor is absent, the change of morphodynamics associated with NREM sleep is largely decreased as well as the correlation between neuronal activity and microglial morphodynamics. This indicates that fractalkine released by neurons could be activity-dependent and constitutes an important signaling mechanism involved in the control of microglial morphodynamics by neurons. Interestingly, Cx3cr1 impairment did not affect

the EEG spectrum nor the quantity of sleep in our mouse model of head restriction. It is important to note that the model we are using to invalidate *Cx3cr1* consists in the insertion of eGFP in the *cx3cr1* gene. This has two consequences. First, in heterozygous mice (*Cx3cr1*[+/-]) that we used for our experiments, the quantity of Cx3cr1 is lower than in wild-type mice, which could impact the physiology of the mouse. Second, in *Cx3cr1*[-/-] mice, the expression of eGFP is twofold when compared to heterozygous mice. This has been taken into account when performing imaging as we reduced the laser power to reach an eGFP signal equivalent to the one measured in heterozygous mice. This increased expression of eGFP should not interfere with the effects we are reporting as we compared morphodynamics between wake and sleep within the same mouse.

Our study, along with several others, has provided evidence that microglial dynamics seems to be activity-dependent and microglial processes are attracted towards sites of increased activity[8,12]. Elucidating neuronal activity during sleep has been an open question for many years. Contrary to Ca²⁺ transients in the soma, when calcium dynamics are monitored at the dendritic level, the decrease in calcium activity was not observed during NREM sleep[47]. In a similar manner, our monitoring of calcium activity in spines indicates that the frequency of Ca²⁺ transients does not change between wake and NREM sleep at the spine level. Furthermore, NREM sleep is composed of different EEG rhythms: slow-wave (0.5–4 Hz) and spindle-rich activity (9–16 Hz), that have profoundly different activity and possible impact on microglial cells. Even though we report decreased microglial morphodynamics during NREM sleep, the impact of the different stages of sleep may not be straightforward and needs to be assessed more closely in future studies using a better temporal resolution. Nonetheless, we found that microglial processes were overall closer to spines and spent more time in contact with spines active during wake (Fig. 6). The positive attraction was affected by NREM sleep as microglial processes were farther from spines and the correlation between spine activity and contact duration was reduced. Our findings suggest that microglial processes seem to sense neuronal activity predominantly during wake, and sleep might hinder this attraction. The exact mechanisms by which microglia-spine interactions differ during sleep and wake warrant further investigation and are beyond the scope of this paper, but a regulation of the expression of Cx3cr1 cannot be excluded as described in ref. 46.

Microglial contact with neuronal components has previously been found to increase local spine activity or homeostatically downregulate neuronal activity[8,9,13]. Like previously reported by Akiyoshi et al., we also observed an increase in spine activity during contact with microglial processes (Fig. 7). The exact mechanisms remain unknown, but microglia may secrete many soluble molecules locally, such as cytokines, neurotrophic factors, and NT, capable of influencing neuronal activity[48]. They may also release extracellular vesicles that may increase glutamate release at presynaptic sites[49] and trigger intracellular Ca²⁺ elevation by contact-dependent mechanism[50]. Unexpectedly, the increase in spine activity during microglial contact took place during sleep, and to our knowledge, this is the first report of microglia modulating spine activity in a specific vigilance state.

Since the function of sleep has been associated with synaptic homeostasis and memory, a tempting speculation would relate these specific contacts with synaptic scaling mechanisms. Two predominant theories of synaptic scaling during sleep are hotly debated. The active system consolidation theory proposes that sleep is involved in strengthening of synapses tagged during wake, by active replay during slow wave sleep[51]. On the other hand, the synaptic homeostasis hypothesis suggests a global synaptic downscaling during sleep, which counters synaptic potentiation generated during wake[52]. Even though microglia may be involved in both functions, our findings suggest that during sleep, microglia may be less oriented toward sensing neuronal

activity, but potentially towards exerting specific functions at synapses, such as strengthening specific synapses.

Furthermore, Akiyoshi et al., suggested that microglia-dependent increase in spine activity may contribute to local network synchronization[8]. Taking into consideration that spindle-rich NREM sleep is associated with increased and synchronized activity in dendrites[47], it is possible that microglia may participate in local dendritic spindle-related plasticity during sleep. Contrary to Akiyoshi et al., a brain-wide ablation of microglia led to increased neuronal synchronization in the striatum[9], suggesting that the microglia-neuron interaction and functional consequences may be region-dependent. Bearing in mind these findings, future studies need to determine the temporal and spatial requirements for increased network synchronization caused by contact-dependent microglial increase in spine activity.

In conclusion, despite being the resident immune cells of the brain, recent studies attribute fundamental physiological functions to microglial cells, including contribution to synaptic connectivity and properties, as well as the regulation of neuronal activity and network. These newly discovered tasks depend on the integrity of microglial morphology and motility. We demonstrate that microglial motility and morphology are modulated during the vigilance states, both globally and at the level of the spine. Spine activity, and especially the vigilance state in which activity is occurring, impacts microglial proximity and contact with spines, resulting in functional consequences during contact. Based on our findings, we propose that microglial processes may sense neuronal activity particularly during wake and increase spine activity during sleep. These findings provide a basis for future work in understanding the mechanisms regulating microglial dynamics and microglia-spine activity across the vigilance states and excitingly, the potential functions of microglia in synaptic homeostasis. Understanding these mechanisms at the physiological level is crucial for understanding how sleep disruptions and microglial activation in pathological conditions may impact these processes.

## Methods

### Animals

Microglial motility and complexity imaging was performed in six to ten-week-old male *Cx3cr1*[+/eGFP] mice that we refer as to *Cx3cr1*[+/-] in the text, that expressed enhanced green fluorescent protein (eGFP) under the control of the Cx3cr1 promoter[53], while *Cx3cr1*[eGFP/eGFP] mice (*Cx3cr1*[-/-] in the text), in which Cx3cr1 was knocked out on both alleles, were used to image microglial morphodynamics when Cx3cr1 signaling is invalidated. For microglial dynamics and spine activity analysis, *Cx3cr1CreERT2*[+/-] mice[6] were crossed with *ROSA26-STOP-tdTomato*[-/-] mice, to generate *Cx3cr1CreERT2*[+/-]*ROSA26-STOP-tdTomato*[-/-] mice. All transgenic mice were 6 to 10-week-old males, derived from the C57BL/6J strain. Mice were housed in enriched and ventilated cages with bedding and running wheels, at $22 \pm 2$ °C under 12/12 h light/dark cycle (light onset at 7:00 a.m.) and were given access to food and water ad libitum at *ALECS, SCAR,* and *ANIPHY Facilities*. All animal procedures were conducted in accordance with the Guidelines of the Animal Care Facility of University Claude Bernard Lyon 1 and were approved by French Ministry of Agriculture (Apafis #DR2014-14, Apafis #7839, Apafis #10350 and Apafis #20983) and Local Ethics Committee CE2A55.

### Induction of Cre activity with Tamoxifen treatment

Tamoxifen (catalog #T-5648; Sigma-Aldrich) was dissolved in warm sterile olive oil (catalog #8001-25-0, Sigma Aldrich, warmed at 55 °C) at a concentration of 20 mg/ml. Six to seven-week-old *Cx3cr1CreERT2*[+/-]*ROSA26-STOP-tdTomato*[-/-] mice were injected subcutaneously in the thigh at 0.4 mg/g, twice, 48 h apart.

## Surgery and habituation

Mice were deeply anesthetized with isoflurane (3–4%, Isovet, Piramal Healthcare, UK Ltd.) and mounted in a stereotaxic frame (D. Kopf Instruments). Isoflurane anesthesia was maintained at concentrations of 1–3% during the surgical procedure. To reduce postoperative pain and inflammation, we administered Carprofen (5 mg/kg s.c.) before surgery, and for two consecutive days following surgery. After the skull was thoroughly cleaned and exposed, two EEG screws were inserted in the frontal and parietal cortex of the right hemisphere and two EMG electrodes were inserted in the dorsal neck muscles. A custom-designed 0.5-mm diameter cranial implant was firmly glued on the left hemisphere using acrylic-based dental adhesive resin cement (Super bond; Sun Medical). The skull was carefully thinned over the somatosensory cortex using a high-speed dental drill until reaching 20–30 μm bone thickness. Most of the drilling was performed in cold, sterile saline solution which allowed continuous cooling and humidification of the bone to avoid heat-induced tissue injury, desiccation of the bone, and inflammation. To visualize $Ca^{2+}$ dynamics in spines of L2/L3 pyramidal neurons of the somatosensory cortex, we performed a small craniotomy (300 μm) and injected 500–700 nl of AAV1.Syn.G-CaMP6m.WPRE.SV40 (UPenn Vector Core, ≥$1 \times 10^{13}$ viral genomes/ml) at a speed of 0.1 μl/min. For injection, we used a glass pipette with a 20 μm diameter tip which was maintained in the brain for 10 additional minutes to avoid backflow. A cover glass was finally set on top of a thin layer of cyanoacrylate glue over the thinned skull.

Following surgery, mice were left to recover for one week. Mice were then subjected to daily head-restrained and experimental environment habituation sessions for imaging. The duration of the training sessions increased progressively from 10 min to 4 h. At the beginning and end of each session, mice were rewarded with several drops of sweetened concentrated milk to associate the head restraint condition with a positive experience.

## Vigilance state recordings

The EEG/EMG was monitored and continuously recorded using a differential amplifier (Model 3000, A-M systems). EEG and EMG signals were sampled at 1 kHz and band-pass filtered with 0.5–300 Hz and 10–500 Hz, respectively, and recorded via Prairie View (Bruker, Nano Surfaces Division, Madison, WI, USA). EEG/EMG data were analyzed using a custom Matlab (Mathworks, Massachusetts, USA) script. EEG power as well as EMG power spectra were calculated over a 4s-width sliding time window. Wake state criteria were: Increased EMG and high theta/delta ratio (>2). NREM sleep criteria were defined by low EMG activity power and high delta/theta ratio (>2), while REM sleep criteria consisted of high theta/delta power ratio (>4) and muscle atonia, characterized by an almost null EMG power. Sleep and wake episodes were defined using a manually established EMG threshold to fit with expert scoring. Episodes containing >90% and <25% of wake time were recognized as wake and sleep respectively.

## Two-photon in vivo imaging

Ultima two-photon laser scanning microscope (Bruker Nano Surfaces Division, Madison, WI, USA) with a mode-locked Ti:Sapphire laser (InsightX3, Spectra-Physics) and 20× water-immersion objective (0.95 N.A. Olympus) were used. Fluorescence was detected using a 560 nm dichroic mirror coupled to 525/50 nm and 650/40 nm emission filters for eGFP and tdTomato, respectively. Laser power during imaging was maintained below 20 mW.

For 3D imaging of microglial cells only, the laser was tuned to 900 nm due to the excitation wavelength for eGFP. Microglial cells were imaged in the somatosensory cortex at a depth of 60–150 μm from the cortical surface. The imaging area was 200 × 200 μm sampled on 512 × 512 pixels i.e., pixel size of 0.38 μm. Z-stacks containing 25–35 consecutive images were acquired every 30 s, with a step size of 1 μm/optical section. A typical recording lasted 30–35 min. All imaging

sessions were performed at the beginning of the light phase, corresponding to the maximum sleep pressure.

3D imaging of microglial dynamics and spine calcium activity was performed using a laser tuned to 980 nm for simultaneous excitation of both fluorescent proteins eGFP and tdTomato. Imaging was performed in the L1 of the somatosensory cortex at a depth of 60–120 μm from the cortical surface because of higher spine density. The imaging area was 100 × 100 μm sampled on 256 × 256 pixels i.e., pixel size of 0.4 μm. Z-stacks containing 14–16 consecutive images were acquired every 7 s, with a step size of 1 μm/optical section. A typical recording lasted 35 min (300 Z-stacks).

For Microglial dynamics and calcium imaging experiments during whisker stimulation, 3D imaging of microglial dynamics was performed using two laser lines: a first laser line tuned to 1040 nm for excitation of the fluorescent protein tdTomato, and a second laser line tuned to 950 nm for excitation of the fluorescent protein eGFP. This configuration was chosen to minimize tdTomato bleaching. Imaging was performed in the L1 of the C2 barrel cortex at a depth of 60–120 μm from the cortical surface because of higher spine density. The imaging area was 100 × 100 μm sampled on 256 × 256 pixels i.e., pixel size of 0.4 μm. A resonant scanner and a 400 μm piezo controlling device allowed us to perform Z-stacks (14 planes) acquisitions every 3.4 s, with a step size of 1 μm/optical section. To partially compensate for the difference in temporal resolution, we acquired four Z-stacks of the activity channel for every Z-stack of the microglia channel. A typical recording lasted ~35 min (650 Z-stacks). Baseline was imaged for 10 min, followed by a period of 25 min during which whisker stimulations were performed.

Whisker stimulations were performed using a piezoelectric blade (PL140.11 PI Ceramic®) driven by a home-made microcontroller device. Stimulations were triggered each 30 s and lasted for 1-s at a frequency of 90 Hz[54]. The C2 whisker was inserted into a glass pipette fixed to the piezo blade, and we made sure that the whisker was still in the glass pipette at the end of each imaging session. Contact durations were measured on contacts happening during the stimulation period. To take into account the contacts which lasted longer than the 20 min period of stimulation we kept the contacts which lasted during all the stimulation period and started before the onset of the stimulations.

## Analysis

Image processing and analysis were performed using custom-written Matlab scripts and LabVIEW. Brightness and contrast, as well as intensity normalization and noise filtering were performed before 3D co-registration calculated by triple Fourier Transform. The microglial cell Z-stacks were then corrected for drift and movements in the $x$, $y$, and $z$ planes during time-lapse image acquisition. Each volume was registered to the first volume chosen as reference. The shift measurement was obtained from volumes' cross-correlation calculated by fast Fourier transform. When animal movements were too important for automated registration, the time points were removed from the analysis.

For analysis of microglial motility and complexity during wake and sleep experiments, we manually delimited and cropped regions of interest containing the totality of one microglial cell from the 200 × 200 μm field of view using ImageJ (National Institute of Mental Health, Bethseda USA). Standard deviation intensity projections from the Z-stacks were performed to generate 2D time-lapse movies. To assess the complexity of microglial cells during wake and sleep, we converted the projections into binary and calculated the Hausdorff fractal dimension[55] for each time point. The Hausdorff dimension is a measure of roughness also called fractal dimension $D$. This measure corresponds to the slope of the log-log characteristic function, drawing the number of tiles of size $\varepsilon$ needed to cover all the surface of the object of interest. To analyze microglial motility, we performed subtractions between two consecutive Z-stack projections. For

visualization purposes, the lost and gained pixels between two consecutive time steps were pseudo-colored in magenta and cyan respectively, resulting in images where magenta and cyan points represented lost and newly formed processes, respectively. The total number of pixels of the absolute difference of two consecutive images determined the global motility coefficient (arbitrary unit).

Due to the arbitrary range of values for these parameters, after subtracting the median of the data for each cell, the cells were normalized for each parameter to fit a scale from +100 to −100. Thus, the data are all in the same range and their dynamics are comparable. Microglial complexity and motility index during wake and sleep were obtained by averaging the normalized values during wake and sleep for each imaging session.

For analysis of microglial dynamics and spine calcium imaging during wake and sleep experiments, we first localized the spines in the xy plane using a maximum intensity projection in time (MIPT). Then, we localized the time point corresponding to the highest intracellular $Ca^{2+}$ level for each spine. We found the Z position containing the three main planes of the spine volume ($3 \mu m$ volume), and performed a Z maximum intensity projection (MIP) time-series of these planes. On the MIP time-series, we delimited a $15 \times 15 \mu m$ region of interest (ROI) centered on the spine, and generated a new time-series containing only the ROI. Then, we constructed a microglia time-series with 7–9 planes with the same ROI size encapsulating the spine of interest. Finally, standard deviation intensity (STD) projections of the microglial cell images were performed. The maximum intensity and standard deviation projections were used for further analysis of spine activity and microglial dynamics respectively. For most of the microglia-spine analyses, we used 11–22 spines/mouse from five mice. For spine activity during sleep and wake, with and before/after contact, we used 10–18 spines/mouse from five mice. Spines that were not in contact with microglial processes during sleep or wake were excluded from analyses ($n = 16$). For microglial proximity with dendritic spines, we used 7-15 spines/mouse from 5 mice, including spines that were never contacted by microglial processes. All spines that were very close to microglial cell bodies or primary processes were excluded from the microglia-spine distance analysis.

To quantify calcium activity in spines, we filtered the signal to remove background noise and measured the fluorescence intensity at the center of the spine and calculated the mean (baseline) and the standard deviation of the fluorescence intensity over time. Significant $Ca^{2+}$ events were defined when the fluorescence intensity exceeded 0.85*std above the baseline. This threshold was determined by visual inspection for several spines allowing us a good detection of the $Ca^{2+}$ transients.

To measure the distance and contact duration between microglial processes and dendritic spines, we used a custom Matlab script that plots the distance between the center of the spine and the moving front of the closest microglial process (Supplementary Fig. 1a). Since the size of the spine is variable, the threshold for microglia-spine contact was defined by visual inspection. Physical contact between microglial processes and dendritic spines was considered when the process was less than 1 pixel away from the edge of the spine and detectable in more than 3 consecutive focal planes. The total contact duration was the cumulative time-step (7 s) during which contact was observed.

To study microglial dynamics with regards to activity during sleep or wake, we classified spines in three categories depending on their level of activity during different vigilance states (Supplementary Fig. 1b): 1) for spines active during wake, the ratio of $Ca^{2+}$ events between wake and sleep was >1.3; this value corresponded to -1*std; 2) for spines mostly active during sleep, the ratio of $Ca^{2+}$ events between sleep and wake was >1.3 and 3) intermediary spines, active both during wake and sleep, with a ratio of $Ca^{2+}$ events <1.3. For illustration purposes, brightness and contrast of the images were adjusted.

For analysis of microglial dynamics and spine calcium imaging during whisker stimulation the image stacks were processed as described above. From three mice in which nine acquisitions were performed, a total of 20–70 spines/acquisition were analyzed, representing a total of 374 spines. All spines that were very close to microglial cell bodies or primary processes were excluded from the microglia-spine distance analysis as it would affect measurements. Likewise, spines selected for the analysis had to be round-shaped of $1–2 \mu m$ in $x$–$y$ and not located on a visible dendrite, regardless of microglial contacts. To quantify calcium activity in spines, we filtered the signal to remove background noise and measured the fluorescence intensity at the center of the spine and calculated the median (baseline) and the standard deviation of the fluorescence intensity over time. Significant $Ca^{2+}$ events were defined when the fluorescence intensity exceeded 0.3*std above the baseline. This threshold was determined by visual inspection for several spines allowing us a good detection of the $Ca^{2+}$ transients. The $Ca^{2+}$ transient response to the stimulations was expressed as the ratio of the area under the curve ($\int \Delta F/F_0 dt$, %s) prior to and after the stimulus onset over 10 min. Individual calcium transients occurring within 6 s following the stimulation were considered as successful evoked responses. The stimulation success rate has been calculated as the ratio of the number of successful stimulations vs the total number of stimulations. A custom Matlab script that calculates the distance between the center of the spine and the moving front of the closest microglial process was used to measure the distance and contact duration between microglial processes and dendritic spines. Since the size of the spine is variable, the threshold for microglia-spine contact was defined by visual inspection. Physical contacts were defined as a distance lower than 1 pixel between microglial processes and dendritic spines in more than 2–3 consecutive focal planes. The total contact duration was the cumulative time step during which contact was observed.

Spines were categorized into three classes based on the change of activity during the stimulation period: 1) Spines for which the $Ca^{2+}$ activity ($\int \Delta F/F_0 dt$) during the 10 min following the stimulation onset is >1.5 fold the baseline activity was considered as spines responding by an increased activity; 2) Spines displaying a ratio <0.6 were considered as decreased activity spines; 3) Spines exhibiting a ratio in [0.6–1.5] range were considered as stable spines. The thresholds for this classification were established in order to have clear responses to stimulation, homogeneous groups, and blind with respect to microglial contacts and durations of contacts.

## Statistics

All statistical analyses were performed using Matlab and the Prism V statistical analysis software (GraphPad, La Jolla, Ca). Microglial complexity and motility during the vigilance states were compared by using the non-parametric Wilcoxon test where n-values represent individual animals. Sleep/wake cycles comparison between heterozygous $Cx3cr1^{+/-}$ and homozygous $Cx3cr1^{-/-}$ mice was obtained using a Mann-Whitney test. For EEG spectra comparison between $Cx3cr1^{+/-}$ and $Cx3cr1^{-/-}$ mice, two-tailed unpaired t-test was performed. For cross-correlation analysis between $Cx3cr1^{+/-}$ and $Cx3cr1^{-/-}$ mice, the peak amplitude $P$ (mean of correlation coefficient at 0, 30, and 60 s) was compared to baseline B (considered from −1000 to −500 s and from 500 to 1000 s as these were considered too far to induce any effects) with Wilcoxon test. For experiments assessing the relationship between microglial dynamics and spine activity, n-values represent individual spines, and we used two-tailed paired t-test or unpaired t-test where appropriate, one-way ANOVA for repeated measures over time, Kolmogorov-Smirnov test for cumulative distributions and Pearson's correlation test for correlation analyses. Grubbs' test was applied to identify outliers, and single outliers were removed from the analysis. All

values reported are mean ± SEM. A significance *p* value threshold *p* < 0.05 was chosen for all analyses.

## Reporting summary

Further information on research design is available in the Nature Research Reporting Summary linked to this article.

## Data availability

Source data are provided with this paper. Raw image files are stored on servers at Université Claude Bernard Lyon 1 owing to their large size. These raw data can be provided from the corresponding author upon request.

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

## Acknowledgements
The authors would like to thank Paul Salin and Luc Gentet for their valuable inputs, advice on protocols and scoring of the vigilance states; Denis Ressnikoff from the CIQLE platform (UCBL1, Lyon) for technical assistance; the *ALECS, SCAR,* and *ANIPHY* animal facilities (Lyon, France) for mouse breeding, housing and injections and Olivier Peyruchaud for sharing the *ROSA26-STOP-tdTomato* mouse line. This research was funded by the French national research agency (ANR) project MICRO-MEM ANR-17-CE16-0008 and LabEx CORTEX. I.H. was supported by a research grant from the *Université Claude Bernard Lyon 1* attributed by the doctoral school *Neurosciences et Cognition* (ED476), K.C. was supported by a research grant MICROMEM ANR-17-CE16-0008 and LabEx CORTEX.

## Author contributions
Conceptualization and experimental design by I.H., K.C., M.R., and O.P. Experiments were conducted by I.H., K.C., and M.R. Data analysis and statistics were performed by I.H., K.C., M.R., and J.C.C. Codes were written by J.-C.C. The manuscript was written by I.H., K.C., M.R., and O.P. with contributions from J.-C.C. and J.H.

## Competing interests
The authors declare no competing interests.
