## [Peer Review File · Nature Communications]

Sleep decreases neuronal activity control of microglial dynamics in miceEditorial Note: This manuscript has been previously reviewed at another journal that is not operating a transparent peer review scheme. This document only contains reviewer comments and rebuttal letters for versions considered at *Nature Communications*.

REVIEWERS' COMMENTS

Reviewer #1 (Remarks to the Author):

Sleep decreases neuronal activity control of microglial dynamics

The authors report that microglia dynamics and contact with neurons are modulated during vigilance states and by neuronal activity. Interesting, they report decreases in overall microglial motility and complexity during periods high EEG power, consistent with NREM sleep. They report that state dependent changes are dependent upon the fractalkine receptor, CX3CR1. The paper is much improved since the last review and all of my recommendations, with the exception of one, were satisfactorily met. I recommended the following minor edits, primarily as to better elucidate the aims of the study and which questions the study seeks to answer, as well as some points of methodological clarification.

Most important points:

- I strongly recommend the authors add 1-2 sentences either in the discussion or methods addressing the possibility that dosing of GFP may lead to differential visualization of smaller processes which may alter the outcomes of the sleep/wake imaging independently of loss of Cx3CR1. It was discussed in the letter but I believe this should be in the main text.
- The authors have forgotten to include figure 4.

Minor Comments:

- I find the beginning of abstract to be little choppy.
- Line 6: In some contexts, there is good evidence to suggest that these contacts have been shown to be activity dependent (PMID: 19339593, PMID: 21164543, PMID: 32999463, and PMID: 31776478; all of which the authors cite), though of course gaps do remain in our knowledge. The authors may consider rewording this sentence, either to make it more specific to their specific context.
- Lines 36-41: The sentence beginning with "On the other hand...." is awkward. I might suggest refining and using "However" instead of "on the other hand." Further, it is unclear what is meant by "the state of microglia." The next sentence is also awkward. I might suggest breaking it into two sentences and stressing the context and vigilance state dependent nature of the study. If I understand correctly, I believe the question the authors wish to stress is not whether or microglia contact synapses in an activity dependent manner, but rather how and in what context, are such microglial contacts modulated.
- Line 43: It is unclear what "these states" refers to.
- Line 176: Space permitting, I might suggest adding a brief clarification as to what a Hausdorff fractal dimension quantification is and how it is applied in this context.
- Lines 381-382: I suggest moving this information to the first mention of microglial imaging and/or to the methods section.
- More a point of curiosity but did the authors observe changes in microglial process distance with absence of CX3CR1 between sleep states?
- In the discussion section, I suggest the authors indicate which figure the results being discussed are from.
- Line 507: The citations the authors provide do not address the effect of the craniotomy preparation on neural firing dynamics or microglial activation (citations 37 and 38). Could the authors provide more evidence for this claim?
- Line 529: "is" should be "are"
- Line 589: I suggest the word "by" be changes to "during".

Additional comments from reviewer 1 in relation to reviewer 3's previous comments:

The authors have addressed most of these concerns. I do agree that the authors have presented a large body of work and that further experiments are not necessary. However, I feel the authors could have done a better job putting their findings in the context of the current literature and discussing the pitfalls of some of their approaches. Statements such as claiming that this is the first study to "assess mammalian microglial in their genuine environment during different vigilance states." seem overblown. I do not think that pharmacological manipulation or whisker stimulation or even genetic ablation makes an environment any less "genuine" than another. This concern aside, the authors conclusions as to the implications of their study are commensurate with their findings. I might suggest amending line 501 to read "Our study is the first to examine microglial morphodynamics during physiological sleep." The use of CX3CR1^{-/-} or rather the CX3CR1-gfp/gfp animals warrants more discussion in the main body of the text as it is a potentially important caveat of the study. Especially so for those wishing to replicate or expand on the results. It would have, for instance, been ideal in this instance to use CX3CR1gfp⁻ to compare to CX3CR1-gfp/+ animals. I don't think more experiments are needed just some more discussion along with what implications haploinsufficiency may have on the experimental findings and how this could be addressed in the future. Lastly, , I do not believe the authors fully addressed the implications the Nature Neuroscience studies on the effects of norepinephrine on microglia. How might the authors reconcile the effect the changes in noradrenergic tone have been shown, or at least suggested to have, on microglial morphodynamics? I agree that there is much still to be learned in this area, but to suggest the discrepancies between those studies and their own was due to the inflammation of cranial window preparation seems to still leave the reader wanting, especially when the papers cited as evidence of such inflammation do not pertain to it (and at least one of the studies uses thin skull preparations as well). I find myself as a reader still asking how might noradrenergic tone play a role in what the authors describe, or at least what the authors believe it might be. The authors do discuss later within the section but their statements of the effect of noradrenergic tone on microglia seem at odds with their own results and with what is known about NE levels in different sleep states (PMID: 19833104). I might also suggest combining the two paragraphs from lines 501-533. The authors switch between discussing of the effects of neurotransmitters and ions in the section and it may be clearer to discuss first one aspect then the other.

Answer to the reviewers

Reviewer #1 (Remarks to the Author):

Sleep decreases neuronal activity control of microglial dynamics The authors report that microglia dynamics and contact with neurons are modulated during vigilance states and by neuronal activity. Interesting, they report decreases in overall microglial motility and complexity during periods high EEG power, consistent with NREM sleep. They report that state dependent changes are dependent upon the fractalkine receptor, CX3CR1. The paper is much improved since the last review and all of my recommendations, with the exception of one, were satisfactorily met. I recommended the following minor edits, primarily as to better elucidate the aims of the study and which questions the study seeks to answer, as well as some points of methodological clarification.

We thank the reviewer for his/her kind comments and for acknowledging the effort made to meet the reviewer's recommendations.

Most important points:

- I strongly recommend the authors add 1-2 sentences either in the discussion or methods addressing the possibility that dosing of GFP may lead to differential visualization of smaller processes which may alter the outcomes of the sleep/wake imaging independently of loss of Cx3CR1. It was discussed in the letter but I believe this should be in the main text.

This has now been added to the discussion: "It is important to note that the model we are using to invalidate Cx3cr1 consists in the insertion of eGFP in the *cx3cr1* gene. This has two consequences. First, in heterozygous mice (Cx3cr1^{+/-}) that we used for our experiments, the quantity of Cx3cr1 is smaller than in wild type mice, this could thus impact the physiology of the mouse. Second, in Cx3cr1^{-/-} mice, the expression of eGFP is twofold when compared to heterozygous mice. This has been taken into account when performing imaging as we reduced the laser power to reach an eGFP signal equivalent to the one measured in heterozygous mice. This increased expression of eGFP should not interfere with the effects we are reporting as we compared morphodynamics between wake and sleep within the same mouse."

- The authors have forgotten to include figure 4.

We apologize for this mistake the figure has now been added

Minor Comments:

- I find the beginning of abstract to be little choppy.

We changed the first two sentences of the abstract to make it smoother

- Line 6: In some contexts, there is good evidence to suggest that these contacts have been shown to be activity dependent (PMID: 19339593, PMID: 21164543, PMID: 32999463, and PMID: 31776478; all of which the authors cite), though of course gaps do remain in our knowledge. The authors may consider rewording this sentence, either to make it more specific to their specific context.

The sentence has been modified as follows: "These contacts have been reported to be activity-dependent, but this has not been thoroughly studied yet, especially in physiological conditions"

- Lines 36-41: The sentence beginning with “On the other hand...” is awkward. I might suggest refining and using “However” instead of “on the other hand.” Further, it is unclear what is meant by “the state of microglia.”

Thanks a lot for these comments, the sentence has been rephrased as follows: “However, decreases in neuronal activity by sensory deprivation, optogenetic and pharmacological inhibition of neuronal activity have yielded inconsistent results on microglial dynamics and microglia-neuron interactions potentially due to the model and methods used and the activation state of microglia”

The next sentence is also awkward. I might suggest breaking it into two sentences and stressing the context and vigilance state dependent nature of the study. If I understand correctly, I believe the question the authors wish to stress is not whether or microglia contact synapses in an activity dependent manner, but rather how and in what context, are such microglial contacts modulated.

We agree that the next sentence was misleading, mainly because of the bad use of the term activation that was not appropriate when referring to neurons. We have now rephrased the sentence which makes it, I hope, easier to understand. In addition, we reformulated the goal of the study as follows to make it clearer: “In awake mice, noradrenergic tone was recently found to suppress microglial process area and surveillance territory when compared to anesthesia 17,18, but a clear description of the microglia process dynamics during sleep and the underlying regulation is lacking.”

- Line 43: It is unclear what “these states” refers to.

“These states” was referring to sleep and wake states, we have added vigilance to clarify this point

- Line 176: Space permitting, I might suggest adding a brief clarification as to what a Hausdorff fractal dimension quantification is and how it is applied in this context.

We have added the following text: “ Hausdorff dimension is a measure of roughness also called fractal dimension D . This measure corresponds to the slope of the log-log characteristic function, drawing the number of tiles of size ε needed to cover all the surface of the object of interest.” We hope this will help the reader to better understand our analysis method.

- Lines 381-382: I suggest moving this information to the first mention of microglial imaging and/or to the methods section.

We moved this information in the methods section “Microglial motility and complexity imaging was performed in six to ten-week-old male $Cx3cr1^{+/eGFP}$ mice that we refer to as $Cx3cr1^{+/-}$ in the text, that expressed enhanced green fluorescent protein (eGFP) under the control of the $Cx3cr1$ promoter ³⁰, while $Cx3cr1^{eGFP/eGFP}$ mice ($Cx3cr1^{-/-}$ in the text), in which $Cx3cr1$ was knocked out on both alleles, were used to image microglial morphodynamics when $Cx3cr1$ signaling is invalidated”

- More a point of curiosity but did the authors observe changes in microglial process distance with absence of CX3CR1 between sleep states?

This is indeed an interesting question but unfortunately we did not perform experiments on $Cx3CR1^{-/-}$ while monitoring neuronal activity. This is certainly something we will do in the near future.

- In the discussion section, I suggest the authors indicate which figure the results being discussed are from.

We thank the reviewer for the remark, and we have done the modifications.

- Line 507: The citations the authors provide do not address the effect of the craniotomy preparation on neural firing dynamics or microglial activation (citations 37 and 38). Could the authors provide more evidence for this claim?

We thank the reviewer for raising this point, the formulation of the sentence was again misleading. It has now been modified and we have added a more appropriate citation Xu et al 2017 in which they state “We show that in vivo imaging of dendritic spine dynamics through an open-skull glass window, but not a thinned-skull window, is associated with high spine turnover and substantial glial activation during the first month after surgery. ». We thus made the following changes: “However, it is important to note that some of these results were obtained from a craniotomy which was found to profoundly affect spine turnover and microglial activation^{37,38, 39”}

- Line 529: “is” should be “are”
This has been corrected.

- Line 589: I suggest the word “by” be changes to “during”.
It has been done.

Additional comments from reviewer 1 in relation to reviewer 3's previous comments: The authors have addressed most of these concerns. I do agree that the authors have presented a large body of work and that further experiments are not necessary.

We thank the reviewer a lot to agree on that matter.

However, I feel the authors could have done a better job putting their findings in the context of the current literature and discussing the pitfalls of some of their approaches. Statements such as claiming that this is the first study to “assess mammalian microglial in their genuine environment during different vigilance states.” seem overblown. I do not think that pharmacological manipulation or whisker stimulation or even genetic ablation makes an environment any less “genuine” than another. This concern aside, the authors conclusions as to the implications of their study are commensurate with their findings. I might suggest amending line 501 to read “Our study is the first to examine microglial morphodynamics during physiological sleep.”

We have changed the text accordingly to the reviewer’s suggestion. However, we did not know that sleep could be non-physiological, may be the reviewer could provide some literature on the matter?

The use of CX3CR1^{-/-} or rather the CX3CR1-gfp/gfp animals warrants more discussion in the main body of the text as it is a potentially important caveat of the study. Especially so for those wishing to replicate or expand on the results. It would have, for instance, been ideal in this instance to use CX3CR1gfp⁻ to compare to CX3CR1-gfp/+ animals. I don’t think more experiments are needed just some more discussion along with what implications haploinsufficiency may have on the experimental findings and how this could be addressed in the future.

We followed the reviewer’s suggestion and added the following section in the discussion “It is important to note that the model we are using to invalidate Cx3cr1 consists in the insertion of eGFP in the *Cx3cr1* gene. This has two consequences. First, in heterozygous mice (Cx3cr1^{+/-}) that we used for our experiments, the quantity of Cx3cr1 is smaller than in wild type mice, this could thus impact the physiology of the mouse. Second, in Cx3cr1^{-/-} mice, the expression of eGFP is twofold when compared to heterozygous mice. This has been taken into account when performing imaging as we reduced the laser power to reach an eGFP signal equivalent to the one measured in heterozygous mice. This increased expression of eGFP should not interfere with the effects we are reporting as we compared morphodynamics between wake and sleep within the same mouse..”

Lastly, I do not believe the authors fully addressed the implications the Nature Neuroscience studies on the effects of norepinephrine on microglia. How might the authors reconcile the effect the changes

in noradrenergic tone have been shown, or at least suggested to have, on microglial morphodynamics? I agree that there is much still to be learned in this area, but to suggest the discrepancies between those studies and their own was due to the inflammation of cranial window preparation seems to still leave the reader wanting, especially when the papers cited as evidence of such inflammation do not pertain to it (and at least one of the studies uses thin skull preparations as well). I find myself as a reader still asking how might noradrenergic tone play a role in what the authors describe, or at least what the authors believe it might be. The authors do discuss later within the section but their statements of the effect of noradrenergic tone on microglia seem at odds with their own results and with what is known about NE levels in different sleep states (PMID: 19833104).

This point has already been the object of intense discussions with the reviewers, and while we are still not sure we can make such a parallel between the noradrenaline studies, we would like to bring to your attention the recent work of Anita Luthi's lab that indicates that noradrenaline levels can be also high during sleep especially during the spindle oscillations. We thus use this reference to enrich the discussion and provide potential explanation of our results as requested. The following sentence has been added: "It has also been shown recently that noradrenergic tone is high during NREM sleep especially during spindle episodes (Osorio-Forero et al 2021). These latest results indicate that it is difficult to speculate on the role of noradrenaline during vigilance state in absence of proper measurements of the noradrenaline tone during the alternation of wake and sleep in the cortex."

I might also suggest combining the two paragraphs from lines 501-533. The authors switch between discussing of the effects of neurotransmitters and ions in the section and it may be clearer to discuss first one aspect then the other.

This last part of the discussion has been modified as suggested by the reviewer.

We would like to acknowledge, one more time the work of the reviewers that really helped to improve the quality of our work.